# Acoustically actuated ultra-compact NEMS magnetoelectric antennas

Tianxiang Nan[1], Hwaider Lin[1], Yuan Gao[1], Alexei Matyushov[1], Guoliang Yu[1], Huaihao Chen[1], Neville Sun[1], Shengjun Wei[1], Zhiguang Wang[1], Menghui Li[1], Xinjun Wang[1], Amine Belkessam[1], Rongdi Guo[1], Brian Chen[1,2], James Zhou[1,3], Zhenyun Qian[1], Yu Hui[1], Matteo Rinaldi[1], Michael E. McConney[4], Brandon M. Howe[4], Zhongqiang Hu[4], John G. Jones[4], Gail J. Brown[4] & Nian Xiang Sun[1]

State-of-the-art compact antennas rely on electromagnetic wave resonance, which leads to antenna sizes that are comparable to the electromagnetic wavelength. As a result, antennas typically have a size greater than one-tenth of the wavelength, and further miniaturization of antennas has been an open challenge for decades. Here we report on acoustically actuated nanomechanical magnetoelectric (ME) antennas with a suspended ferromagnetic/piezoelectric thin-film heterostructure. These ME antennas receive and transmit electromagnetic waves through the ME effect at their acoustic resonance frequencies. The bulk acoustic waves in ME antennas stimulate magnetization oscillations of the ferromagnetic thin film, which results in the radiation of electromagnetic waves. Vice versa, these antennas sense the magnetic fields of electromagnetic waves, giving a piezoelectric voltage output. The ME antennas (with sizes as small as one-thousandth of a wavelength) demonstrates 1–2 orders of magnitude miniaturization over state-of-the-art compact antennas without performance degradation. These ME antennas have potential implications for portable wireless communication systems.

[1] W.M. Keck Laboratory for Integrated Ferroics, and Department of Electrical and Computer Engineering, Northeastern University, Boston, MA 02115, USA. [2] Westford Academy, Westford, MA 01886, USA. [3] Andover High School, Andover, MA 01810, USA. [4] Materials and Manufacturing Directorate, Air Force Research Laboratory, Wright-Patterson Air Force Base, Dayton, OH 45433, USA. Tianxiang Nan and Hwaider Lin contributed equally to this work. Correspondence and requests for materials should be addressed to N.X.S. (email: nian@ece.neu.edu)

Antennas that interconvert between alternating electric currents and electromagnetic (EM) wave radiation, act as an omnipresent critical component in smart phones, tablets, radio frequency identification systems, radars, etc. One of the key challenges on state-of-the-art antennas lies in their size miniaturization[1–6]. Compact antennas rely on an EM wave resonance, and therefore typically have a size of more than $\lambda_0/10$, that is one-tenth of the EM wavelength $\lambda_0$. The limitation on antenna size miniaturization has made it very challenging to achieve compact antennas and antenna arrays, particularly at very-high frequency (VHF, 30–300 MHz) and ultra-high frequency (UHF, 0.3–3 GHz) with large $\lambda_0$, thus putting severe constraints on wireless communication systems and radars on mobile platforms[4]. New antenna concepts need to be investigated with novel EM waves radiation and reception mechanisms for the reduction of antenna size.

On the other hand, strong strain-mediated magnetoelectric (ME) coupling in magnetic/piezoelectric heterostructures has been recently demonstrated which enables efficient energy transfer between magnetism and electricity[7–17]. The strong ME coupling, if realized dynamically at radio frequencies (RF) in ME heterostructures, could enable voltage induced RF magnetic currents that radiate EM waves, and acoustically actuated nanoscale ME antennas with a new receiving and transmitting mechanism, for EM waves. This concept has recently been theoretically proposed[18, 19]. However, despite of the moderate interaction between the surface acoustic wave and magnetization[20–22], strong ME effect has only been demonstrated at kHz frequencies, or in a static or quasi-static process[23, 24]. Here one question naturally arises: Is it possible to realize efficient energy coupling between bulk acoustic waves and EM waves in ME heterostructures at RF frequencies through ME coupling? Based on our results in this work, we can answer this question affirmatively.

Here we demonstrate the nanoelectromechanical system (NEMS) antennas operating at VHF and UHF frequencies based on the strong ME coupling between EM and bulk acoustic waves in the resonant ME heterostructures (ferromagnetic/piezoelectric). These ME antennas have realized acoustic transmitting and receiving mechanisms in nanoplate resonators (NPR) and

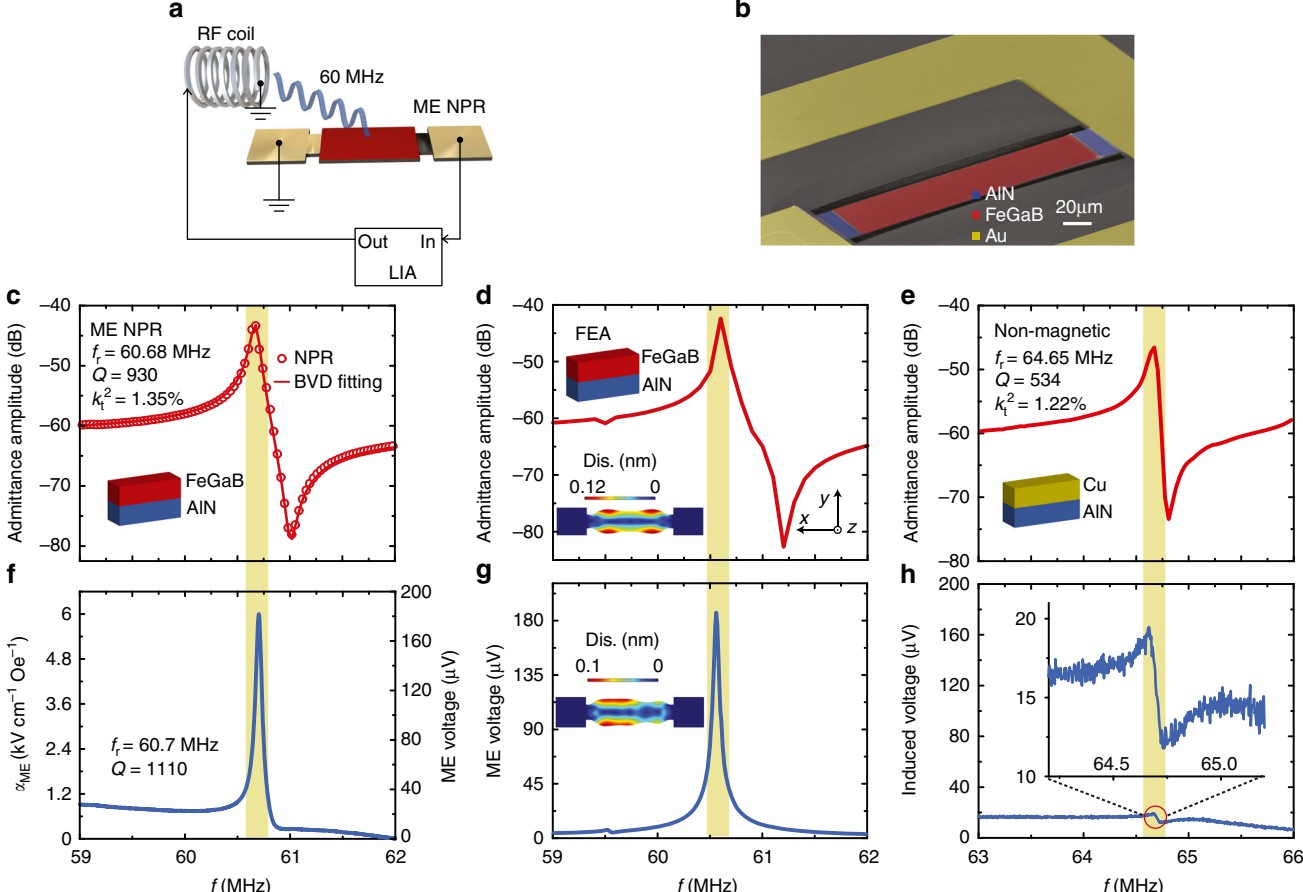

**Fig. 1** ME NPR device with gigantic ME coupling. **a** Schematic representation of the magnetoelectric (ME) nanoplate resonator (NPR) and the induced ME voltage measurement setup by using a high-frequency lock-in amplifier (HFLIA). The RF magnetic field ($H_{RF}$) is generated by a RF coil. **b** Scanning electron microscopy (SEM) images of the fabricated the ME NPR. The red and blue areas show the suspended single nanoplate with FeGaB/AlN ME heterostructure and AlN anchors. The yellow area presents the RF ground-signal-ground electrode. **c** Admittance curve and Butterworth–van Dyke model fitting of the ME NPR with a resonance frequency ($f_r$) of 60.68 MHz, quality factor ($Q$) of 930 and electromechanical coupling coefficient ($k_t^2$) of 1.35%. The inset shows the schematic of the cross-section of the ME heterostructure. **d** Finite element analysis (FEA) of ME NPR for the admittance amplitude. The inset shows the in-plane displacement of the nanoplate at resonance peak position and its coordinate system. **e** Admittance curve of a non-magnetic control sample which has a same device design as ME NPR. The inset shows the schematic of the device cross-section. **f**, ME coupling coefficient ($\alpha_{ME}$) (*left axis*) and the induced ME voltage (*right axis*) versus the frequency of $H_{RF}$ excitation. **g** FEA of ME NPR for the induced ME voltage. The inset shows the in-plane displacement excited by $H_{RF}$. **h** Induced voltage versus the frequency of $H_{RF}$ excitation for the non-magnetic device. The inset illustrates the zoomed-in view of the resonance peak area (*red circle*)

thin-film bulk acoustic wave resonators (FBAR). During the receiving process, the magnetic layer of ME antennas senses $H$-components of EM waves, which induces a oscillating strain and a piezoelectric voltage output at the electromechanical resonance frequency. Conversely, during the transmitting process, the ME antennas produces an oscillating mechanical strain under an alternating voltage input, which mechanically excites the magnetic layer and induce a magnetization oscillation, or a magnetic current, that radiates EM waves. Therefore, these ME antennas operate at their acoustic resonance instead of, EM resonance. Since the acoustic wavelength is around five orders of magnitude shorter than the EM wavelength at the same frequency, these ME antennas are expected to have sizes comparable to the acoustic wavelength, thus leading to orders of magnitude reduced antenna size compared to state-of-the-art compact antennas.

## Results

**Large ME coupling coefficient in the NPR device.** The resonant bodies of the NEMS ME resonators were a 500 nm AlN thin film supporting a $[Fe_7Ga_2B_1 (45 nm)/Al_2O_3 (5 nm)] \times 10$ (hereafter termed FeGaB) thin-film ME heterostructures fully suspended on a Si substrate, where AlN and FeGaB (see Supplementary Note 1 for magnetic properties characterization) serve as the piezoelectric and magnetostrictive element of the ME heterostructure, respectively. The use of a NEMS resonator with an ultra-thin (thickness, $T = 500$ nm) AlN thin film enables efficient on-chip acoustic transduction with ultra-low energy dissipation[25, 26]. In this work, the demonstrated ME antennas span a wide range of frequencies from 60 MHz to 2.5 GHz, which are realized by a geometric design of resonating plates that exhibit different mode of vibrations (Supplementary Note 6).

The strong ME coupling at VHF frequencies was demonstrated through a ME NPR with an in-plane contour mode of vibration (by means of $d_{31}$ piezoelectric coefficient)[27]. In particular, a perpendicular electric field on the piezoelectric AlN layer induces actuation in the plane of the device. Figure 1a presents the schematic of the measurements and the structure of ME NPR which has a rectangular resonating plate consisting of a single-finger bottom Pt electrode and a thin-film FeGaB/AlN heterostructure. All the NEMS ME resonators in this work were fabricated using CMOS (complementary metal-oxide-semiconductor) compatible microfabrication processes (see Method and Supplementary Note 2). The scanning electron microscopy (SEM) image of the NPR ME resonator is shown in Fig. 1b. The length ($L$) and width ($W$) of the FeGaB/AlN active resonant body are 200 and 50 μm, respectively. The ME nanoplate FeGaB/AlN is fully released from the Si substrate but mechanically supported and electrically contacted by the two AlN/Pt anchors for optimized ME coupling with a minimum substrate clamping effect. To study the electromechanical properties of the ME NPR, the electrical admittance curve was characterized by using a network analyzer, as shown in Fig. 1c. The admittance spectrum at resonance can be fitted to the Butterworth–van Dyke model[27], which yields an electromechanical resonance frequency ($f_{r,NPR}$) of 60.68 MHz, a high-quality factor ($Q$) of 930 and electromechanical coupling coefficient ($k_t^2$) of 1.35% indicating a high electromechanical transduction efficiency and low loss (Supplementary Note 3). This $f_{r,NPR}$ corresponds to the contour mode of vibration excited in AlN, which can be analytically expressed as $f_{r,NPR} \propto \frac{1}{2W_0}\sqrt{\frac{E}{\rho}}$, where $W_0$ is the width of the resonator pitch, $E$ and $\rho$ are the equivalent Young's modulus and equivalent density of the FeGaB/AlN resonator, respectively[28, 29]. Finite element analysis (FEA) on the admittance curve of the device with the same geometry is shown in the Fig. 1d, which is in good agreement with Fig. 1c. At the

resonance frequency of 60.56 MHz, the in-plane displacement distribution shown in Fig. 1d inset indicates a contour extensional mode of vibration, in which the bulk of the device structure expands in its plane. It is also notable that the $Q$-factor of this ME resonator is much higher than the conventional low frequency ME heterostructures in previous reports[10, 30–33].

Under the excitation of $H_{RF}$ with an amplitude about 60 nT (provided by a RF coil along the length direction of the resonator, see Supplementary Note 4), the induced ME voltage output of the NPR device was measured by using an UHF lock-in amplifier (UHFLI), as shown in Fig. 1f. A clear resonance peak is shown in the ME voltage spectrum at 60.7 MHz with a peak amplitude ($U$) of 180 μV. The amplitude of the peak is very sensitive to the excitation frequency exhibiting a $Q$-factor that is similar to the admittance curve in Fig. 1c. The experimentally measured output ME voltage spectrum (Fig. 1f) agrees well with the FEA results of the ME voltage spectrum with a peak amplitude of 196 μV as shown in Fig. 1g (Method). Figure 1g inset shows the simulated in-plane displacement of the ME resonator excited by the $H_{rf}$ at its resonance frequency, indicating a contour mode of vibration. The same mode of vibration excited by magnetic field and electric field demonstrates that the strain-mediated ME coupling is dominating. A high ME coupling coefficient of $\alpha_{ME} = \partial U/(\partial H_{rf} \cdot T) = 6$ kV Oe$^{-1}$ cm$^{-1}$ can be obtained at the $f_{r,NPR}$, where[23, 34]. It is notable this ME coupling coefficient is obtained without any DC bias magnetic field, and the value is comparable to recent reported values with optimum bias magnetic field at much lower electromechanical resonance frequencies of kHz[35].

As a comparison, a non-magnetic single-finger NPR has also been tested as a control sample to confirm that the strain-mediated ME coupling is responsible for the observed voltage output under the $H_{RF}$ excitation. For the non-magnetic resonator, a Cu thin film of 500 nm was deposited on AlN plate (Fig. 1e inset) to replace the ferromagnetic FeGaB layer as the top electrode. As shown in Fig. 1e, the Cu/AlN based NPR exhibits a similar admittance behavior (both $f_r$ and $Q$) as the ME NPR (Fig. 1c). Figure 1h shows the $H_{RF}$ induced voltage spectrum of the resonator with a Cu/AlN heterostructure. With the same $H_{RF}$ excitation as the ME resonator ($H_{rf} = 60$ nT), the induced voltage of the Cu/AlN resonator at its electromechanical resonance frequency of 64.7 MHz is very low, about two orders of magnitude smaller than the induced voltage in the FeGaB/AlN ME NPR (Fig. 1c). Note that the induced voltage spectrum profile of the Cu/AlN NPR is highly antisymmetric near its resonance frequency, which is totally different from the symmetric ME voltage spectrum (Fig. 1f) but similar to its admittance spectrum (Fig. 1e). This antisymmetric line shape can be attributed to a weak inductive coupling effect between the device ground loop and EM wave, which could also exist in the FeGaB/AlN NPR device. However, the symmetric ME voltage spectrum in the FeGaB/AlN NPR indicates that the inductive coupling effect has an extremely low efficiency compared to the ME coupling. Thus, the strong resonance peak induced by the $H_{RF}$ in FeGaB/AlN NPR device is resulted from the presence of the ME coupling, in which FeGaB films with high-permeability[36, 37] couples to RF excitation magnetic field very effectively.

The ME NPR with multi-finger interdigitated electrodes, which we have demonstrated recently[17], were found to have negligibly small ME voltage in the same measurement setup, that is over three orders of magnitude smaller at the electromechanical resonance compared to the single-plate ME NPR. This phenomenon has been confirmed through COMSOL simulations (Supplementary Note 5). Single-finger ME resonators produces high ME output voltage as the uniform RF excitation magnetic fields can couple strongly to single nanoplate. While the negligibly ME voltage output in multi-finger ME resonators is

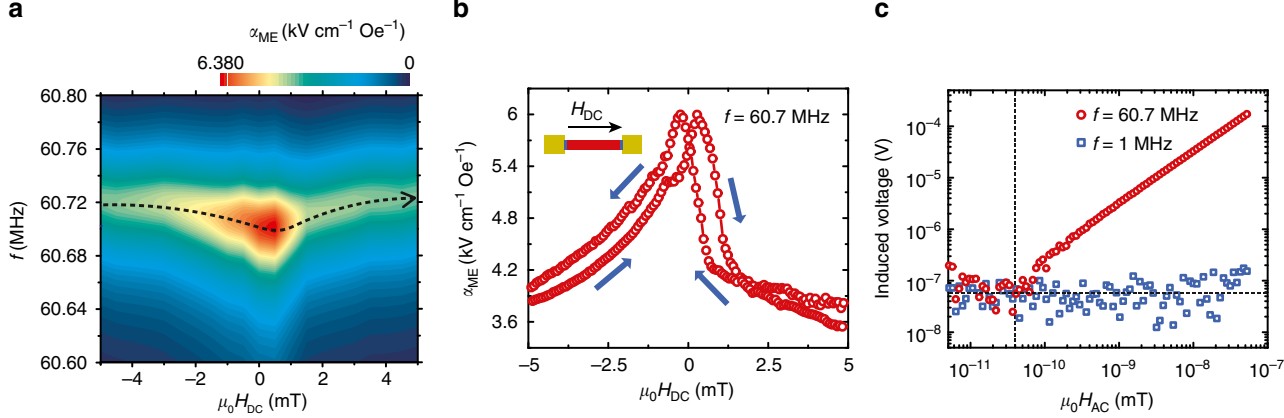

**Fig. 2** Bias magnetic field and frequency dependence. **a** Magnetoelectric (ME) coupling coefficient $\alpha_{ME}$ of ME NPR as functions of DC bias magnetic field $H_{DC}$(x-axis) and the frequency of RF magnetic field (y-axis). The dashed curve exhibits the resonance frequency (highest intensity at each frequency sweep) versus the bias magnetic field. The bias magnetic field was swept from −5 to 5 mT. **b** The hysteresis loop of $\alpha_{ME}$ obtained by sweeping the magnetic field back and force at $f = 60.7$ MHz. The inset shows the schematic representation of the ME NPR with the external bias magnetic field applied along its length direction. **c** Induced ME voltage as a function of magnetic field at excitation frequency of 60.7 MHz (*red*) and 1 MHz (*blue*)

due to the fact that, the uniform $H_{RF}$ do not couple efficiently to the multi-finger NPRs which produce nonuniform RF strain fields and nonuniform magnetization fields.

We further gain insight into the magnetization dependence of the single-finger ME NPR shown in Fig. 1 by examining its ME coupling strength at different bias magnetic fields. The induced ME voltage spectrum was measured with DC bias magnetic fields swept from −5 to 5 mT along the resonator length direction (as shown in the inset of Fig. 2b). Figure 2a shows the $\alpha_{ME}$ as a function of the DC bias magnetic field $H_{DC}$ and the frequency of $H_{RF}$. At zero bias magnetic field $\mu_0 H_{DC} = 0$, the $\alpha_{ME}$ is maximized at the $f_{r,NPR}$ of 60.7 MHz, which is in good agreement with Fig. 1f. At $\mu_0 H_{DC} = \pm 5$ mT, $f_{r,NPR}$ is shifted to 60.72 MHz as shown in the dashed curve of Fig. 2a. This can be attributed to the $\Delta E$ effect[17], that is the bias magnetic field modifies the Young's modulus of FeGaB and thus leads to varied $f_{r,NPR}$ of the resonator[17, 31, 38]. Moreover, a hysteretic behavior of the $\alpha_{ME}$ (at $f_{r,NPR}$) was observed by sweeping the DC magnetic field back and force, with the maximum value of 6 kV cm$^{-1}$ Oe$^{-1}$ at $\pm 0.5$ mT (Fig. 2b). This is consistent with the strain-mediated ME coupling mechanism and the magnetic hysteresis of the FeGaB/AlN nanoplate (Supplementary Note 1). The magnetic field dependence of $\alpha_{ME}$ in the ME NPR provides another direct evidence that the observed interaction between EM wave and acoustic resonance results from the ME coupling.

It is important to note that the strong $\alpha_{ME}$ at zero bias magnetic field directly leads to robust self-biased ME sensors. This is drastically different from conventional ME heterostructures with electromechanical resonance frequencies in the kilohertz frequency range, which show near zero ME coupling at zero bias magnetic field[32, 39–41]. This difference can be attributed to the edge curling wall[42, 43] under self-bias condition for the magnetic/non-magnetic multilayers (FeGaB/Al$_2$O$_3$) used as the magnetostrictive layer in ME antennas. The detection limit of the NPR ME antennas for sensing weak $H_{RF}$ under zero bias magnetic field was also characterized as shown in Fig. 2c, where the induced voltage is plotted as a function of $H_{RF}$ at two different excitation frequencies. At the resonance frequency of 60.7 MHz (red), the linear curve scatters at 40 pT with a limit detection voltage of 0.1 μV, indicating a detection limit of 40 pT for the NPR ME sensor. While at the off-resonance frequency of 1 MHz (blue), the induced voltage randomly distributes around the 0.1 μV, showing no sensitivity to 1 MHz magnetic field excitation with the amplitude of 10$^{-11}$–10$^{-7}$ T.

It is notable that ME NPR antenna arrays with multiple frequency bands from MHz to GHz can be integrated in one wafer by designing the ME NPR with different lateral dimensions (or $W$), since the $f_{r,NPR}$ is inversely proportional to $W^{27}$. This allows the broadband ME NPR antenna arrays on the same wafer, which compensates for the narrowband operation frequencies of ME antennas. The resonance frequencies as well as the $Q$-factors of various NPR (including FBAR) devices fabricated on one wafer are summarized in Supplementary Note 6 as a function of $W$.

**FBAR ME antennas.** We further designed, fabricated, and tested ME antennas that operate at GHz based on the thickness resonance mode of FeGaB/AlN thin-film FBAR devices. The antenna radiation property of the ME FBAR based antennas was tested in a far-field configuration at GHz range in an anechoic chamber. As shown in Fig. 3a, b, the active element of ME FBAR antenna is a suspended FeGaB/AlN ME circular disk with a diameter of 200 μm. This FBAR ME antenna exhibits a thickness extensional mode of vibration as shown in the schematic representation of Fig. 3a. A calibrated linear polarization standard horn antenna and a ME FBAR based antenna are connected to the port 1 and port 2 of a network analyzer, respectively for antenna gain measurements (see Methods). Different from ME NPR, the electromechanical resonance frequency of the ME FBAR ($f_{r,FBAR}$) is defined by the thickness of the circular resonating disk and can be expressed by $f_{r,FBAR} \propto \frac{1}{2T}\sqrt{\frac{E}{\rho}}$. The $f_{r,FBAR}$ was found to be 2.53 GHz by measuring the reflection coefficient ($S_{22}$) of the FBAR device as shown in Fig. 3c, which also exhibits a peak return loss of 10.26 dB and a $Q$-factor of 632. Figure 3c inset shows the simulated out-of-plane displacement of the FBAR indicating a thickness extensional mode of vibration (see Supplementary Note 7 for the simulated $S_{22}$). The receiving and transmitting behavior of ME antennas corresponds to the $S_{21}$ and $S_{12}$ parameters, respectively, as shown in Fig. 3d. Clearly $S_{12}$ and $S_{21}$ curves nearly overlap with each other. These S-parameters ($S_{21}$, $S_{12}$ and $S_{22}$) for the ME FBAR were obtained at zero bias magnetic field for the ME FBAR. The antenna gain for the ME FBAR is measured to be −18 dBi at $f_{r,FBAR}$ through gain comparison method (Methods). It is not trivial to simulate the ME antenna radiation in the framework of a three-dimensional (3D) device. While by using a 1D model, one may not be able to capture the real physics which contain many boundary conditions and anisotropic materials parameters. For example, the magnetic FeGaB

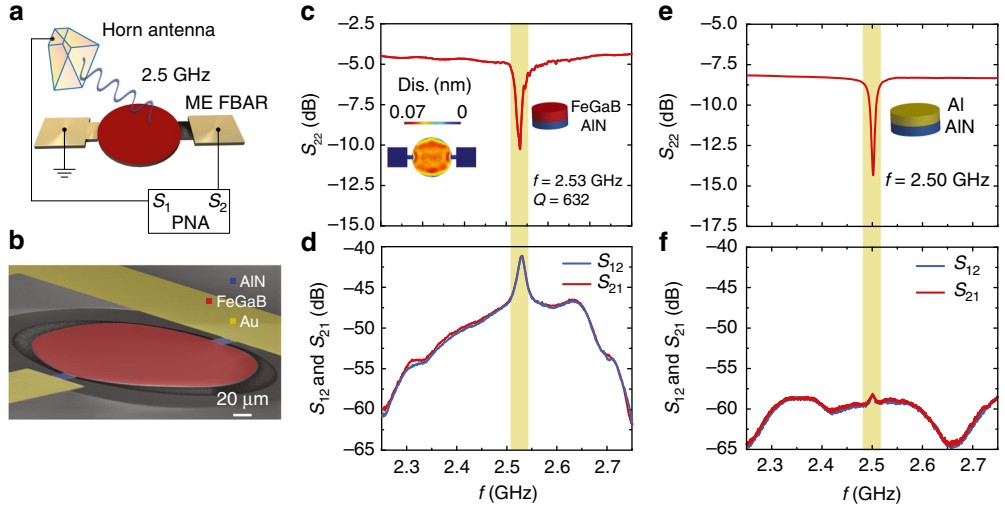

**Fig. 3** ME FBAR antenna. **a** Schematic illustration of the magnetoelectric (ME) thin-film bulk acoustic wave resonators (FBAR) and the antenna measurement setup. The horn antenna and ME FBAR are connected to the $S_1$ and $S_2$ port of a network analyzer. **b** Scanning electron microscopy (SEM) images of the fabricated the ME FBAR. The *red* and *blue* areas show the suspended circular plate and AlN anchors. The *yellow* area presents the electrode. **c** Return loss curve ($S_{22}$) of ME FBAR. The inset shows the out-of-plane displacement of the circular disk at resonance peak position. **d** Transmission and receiving behavior ($S_{12}$ and $S_{21}$) of ME FBAR. **e** Return loss ($S_{22}$) curve of the non-magnetic Al/AlN control FBAR. **f** Transmission and receiving behavior ($S_{12}$ and $S_{21}$) of the non-magnetic Al/AlN control FBAR

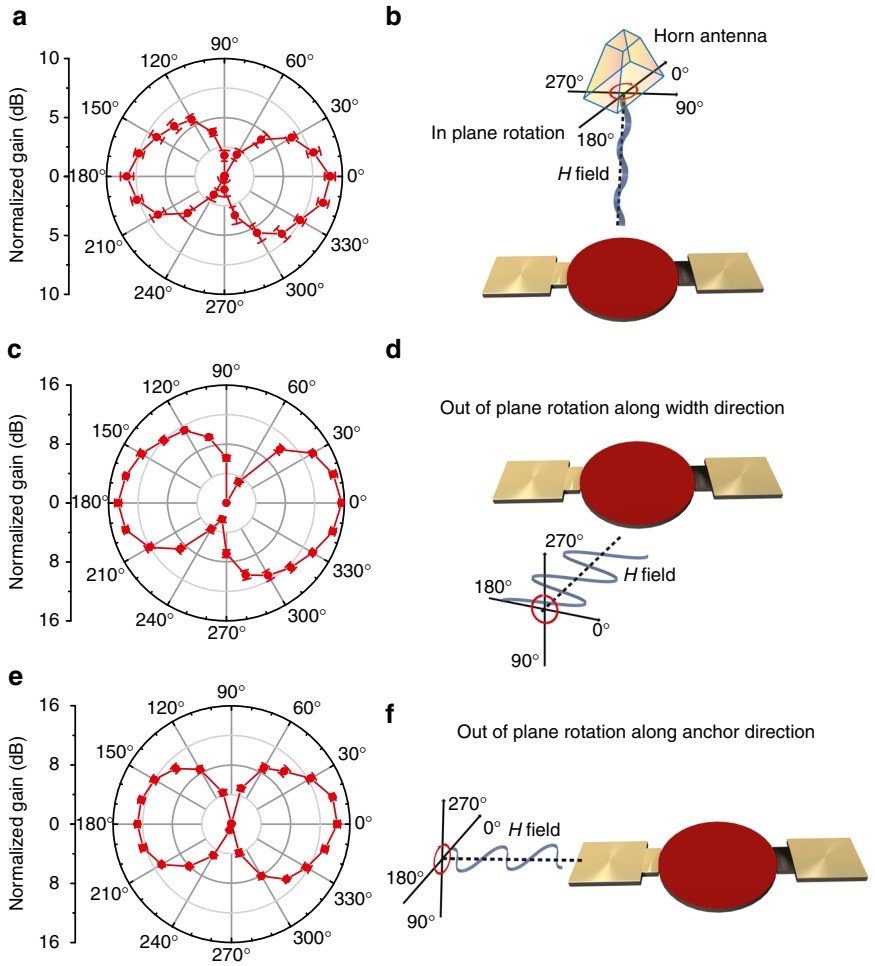

**Fig. 4** ME FBAR antenna measurements under a rotating linearly polarized standard antenna along three major rotational axes. **a**, **c**, **e** Antenna polar gain charts for the in-plane rotation along the width direction, out-of-plane rotation along the width direction **c**, and out-of-plane rotation along the anchor direction **e**. **b**, **d**, **f**. Schematic representations of the experimental setups for **a**, **c** and **e** respectively. The sinusoidal wave along 0° or 180° direction denotes the propagating *H*-field component of the incoming electromagnetic waves

layer in the ME antenna shows a highly anisotropic Young's modulus with a $\Delta E$ effect of 160 GPa along the in-plane magnetic hard axis direction, which is very hard to incorporate into any existing model.

A non-magnetic control device with 1000 nm Al/500 nm AlN has also been tested with the same experimental setups in order to rule out any artificial EM coupling to the ground loop of devices. In the non-magnetic control device, 1000 nm Al was used to replace the 500 nm thick FeGaB multilayer for achieving a device resonance frequency near 2.5 GHz. The loss mechanism of ME antennas is dominated by the mechanical resistance $R_m$ related to the different mechanical damping mechanisms of the magnetic and piezoelectric phases, which is much larger than the radiation resistance $R_r$. The impedance matching is therefore dominated by $R_m$, not $R_r$. Therefore, impedance matching is no longer directly related to the radiation efficiency of ME antennas, which is different from conventional antennas. As shown in Fig. 3e, the Al/AlN control device exhibits a similar electromechanical property as FeGaB/AlN FBAR with similar $S_{22}$ but better impedance matching with an electromechanical resonance frequency of 2.50 GHz. However, no evident $S_{21}$ and $S_{12}$ resonance peak can be observed in the horn antenna measurements in Fig. 3f, except a very weak peak at 2.50 GHz with a peak amplitude just above the noise level, similar to the Cu/AlN NPR control sample shown in Fig. 1h. This suggests that the ME coupling effect dominates in the $S_{21}$ and $S_{12}$ measurement of the ME FBAR antenna.

The radiation behaviors of ME FBAR antenna was also tested by rotating the linearly polarized standard antenna as shown in Fig. 4. The standard antenna can be rotated along one of the three major axes of the ME antenna, the out-of-plane direction (Fig. 4a, b), the in-plane perpendicular to the ME antenna anchor direction (Fig. 4c, d) and the in-plane along the ME antenna anchor direction (Fig. 4e, f). In all the schematics of Fig. 4, the sinusoidal wave along 0° (or 180°) direction denotes the propagating $H$-field component of the incoming EM wave. All three polar gain charts in Fig. 4a, c, e show the similar shape of a sideways figure eight due to the magnetic anisotropy of the FeGaB/Al$_2$O$_3$ multilayer in the circular resonating disk of the ME FBAR. As shown in Fig. 4a, the ME FBAR antenna has the highest gain when the $H_{rf}$ is perpendicular to the anchor direction of the antenna, and lowest gain when the $H_{rf}$ is parallel to the anchor direction. This is because the in-plane magnetic anisotropy of the FeGaB in the circular disk of the FBAR is along the width direction of the ME antenna, and the highest permeability and therefore strongest coupling between $H_{rf}$ and ME antenna is achieved along 0 or 180° direction in Fig. 4a. The other two rotation test configurations in Fig. 4c, e show similar behavior, in which the antenna gain shows its maximum value at 0° (or 180°).This is related to the shape anisotropy of the thin ferromagnetic layer. All the rotational antenna gain measurements at different configurations demonstrate that the high ME antenna gain originates from the strong magnetic coupling between the magnetic field component of the EM wave and the FeGaB of the FeGaB/AlN heterostructure in ME FBAR antennas.

## Discussion

The mechanism for the ME antenna operation and miniaturization is due to the ME effect at the acoustic resonance or electromechanical resonance. Since the acoustic wavelength is much less that of the EM wave resonance, these ME antennas are much smaller than state-of-the-art compact antennas. Size miniaturization of ME FBAR antennas is not due to the high permeability or high permittivity of the ME antennas, which is different from conventional magnetodielectric antenna

approaches. The loss mechanism of ME antennas is also quite different from conventional antennas as the mechanical resistance is dominating the loss of ME antennas. And the mechanical resistance is not directly related to the loss tangent of the piezomagnetic or piezoelectric phases of the ME antennas.

The active area of ME FBAR antenna with a resonating ME resonating circular disk discussed above has a diameter of 200 μm or $\lambda_0/593$, which is 1–2 orders of magnitude smaller than state-of-the-art compact antennas with their sizes over $\lambda_0/10$[1]. As a comparison, the simulated small loop antenna with the same size as the ground loop of the FBAR ME antenna, shows a resonance frequency $f_{r,loop}$ of 34 GHz (see Supplementary Note 8), and the gain of −68.4 dBi at 2.53 GHz due mainly to the poor impedance match, which is 50 dB lower than that of the same size FBAR ME antenna. Clearly these miniaturized ME antennas have drastically enhanced antenna gain at small size owing to the acoustically actuated ME effect based receiving/transmitting mechanisms at RF frequencies. We note that the demonstrated ME antennas are pure passive devices, no impedance matching circuit, or an external power source was used during the measurement. And its maximum achievable bandwidth is within Chu–Harrington limit (Method)[44].

In conclusion, we have demonstrated ME antennas based on NPR and FBAR structures with an acoustically actuated receiving and transmitting mechanism, which are one to two orders of magnitude smaller than state-of-the-art compact antennas. These ME antennas are designed to have different modes of vibration for realizing both VHF (60 MHz) and UHF (2.525 GHz) operation frequencies. Moreover, both NPR and FBAR based antennas can be fabricated on the same Si wafer with the same microfabrication process, which allows for the integration of broadband ME antenna arrays from tens of MHz (NPR with large $W$) to tens of GHz (FBAR with thinner AlN thickness) on one chip by the geometric design of device resonant bodies (Supplementary Note 6). A bank of multi-frequency MEMS resonators can be connected to a CMOS oscillator circuit for the realization of reconfigurable antennas[45]. These ultra-compact ME antennas are expected to have great impacts on our future antennas and communication systems for internet of things, wearable antennas, bio-implantable and bio-injectable antennas, smart phones, wireless communication systems, etc.

## Methods

**Device fabrication**. High resistivity silicon (Si) wafers (>10,000 Ohm cm) were used as substrates for all ME antenna devices. A 50-nm-thick Pt film was sputter-deposited and patterned by lift-off on top of the Si substrate to define the bottom electrodes. Then, the 500 nm AlN film was sputter-deposited, and the via holes was formed by H$_3$PO$_4$ etching to access the bottom electrodes. After that, the AlN film was etched by inductively coupled plasma etching in Cl$_2$-based chemistry to define the shape of the resonant nanoplate. Next, a 100-nm-thick gold (Au) film was evaporated and patterned to form the top ground. Finally, 500-nm-thick FeGaB/Al$_2$O$_3$ multilayer layer was deposited by a magnetron sputtering and patterned by lift-off process. A 100 Oe *in situ* magnetic field bias was applied during the magnetron deposition along the width direction of the device to pre-orient the magnetic domains. Then, the structure was released by XeF$_2$ isotropic etching of the Silicon substrate. The details of the fabrication processes and FBAR antenna layout can be found in Supplementary Note 2.

**Magnetic multilayer deposition**. The magnetic multilayer with the structure of [FeGaB (45 nm)/Al$_2$O$_3$ (5 nm)] × 10 was sputter-deposited on AlN thin film with a 5 nm Ta seed layer at the Ar atmosphere of 3 mTorr with a background pressure of less than $1 \times 10^{-7}$ Torr. The Ta seed layer promoted the FeGaB thin-film growth exhibiting narrow resonance linewidth and close-to-bulk magnetic moment. The FeGaB layer was co-sputtered from FeGa (DC sputtering) and B (RF sputtering) targets. The Al$_2$O$_3$ layer was deposited by RF sputtering using an Al$_2$O$_3$ target. The deposition rates are calibrated with X-ray reflectivity.

**Admittance amplitude measurement**. The admittance curve of resonators was characterized by using a network analyzer (Agilent PNA 8350b). The short-open-load calibration was performed prior to the device measurements. The

transmission parameter $S_{11}$ was acquired and converted to admittance amplitude. The available power at the network analyzer port was set to −12 dBm, and the IF bandwidth was 50 Hz. The devices were tested in a RF probe station with a probe with ground-signal-ground configuration.

**ME voltage measurement**. The induced ME voltage of the NPR was measured by using an UHF lock-in amplifier (UHFLI). The reference current signal was sent to an RF coil to generate a RF magnetic field $H_{rf}$, which has a magnetic field strength simulated by the finite element method. The RF coil is placed 14 mm away from the device under test (see Supplementary Note 4 for the space distribution of $H_{RF}$). The induced ME voltage spectrum is obtained by sweeping the reference frequency (frequency of $H_{rf}$). The ME voltage spectral were also measured under various DC magnetic field.

**Finite element analysis of electromechanical and magnetoelectrical properties**. To analyze the response of the ME structures, the coupling between the magnetic, elastic and electric field in the magnetostrictive and piezoelectric heterostucture are taken into account. Simulations with FEM software, COMSOL Multiphysics V5.1, were carried out to investigate the frequency response. The simulation modules include the magnetic fields, solid mechanics and electrostatics modules. The ME composites were constructed into magnetostrictive, piezoelectric phase and air sub-domain. The simulation were performed at the frequency domain in a 3D geometry. The detail of the analysis can be found in Supplementary Note 9. The linear mechanical, electrical and magnetic parameters of the materials used in this work can be found in Supplementary Note 10. In the demonstration of NPR, we excited the device with an RF magnetic field and used a magnetostatic approximation to simulate the induced voltage in COMSOL. In the FBAR section, we demonstrated the resonance mode and displacement instead of the magnetization dynamics.

**Antenna gain calibration and calculation**. The antenna gain $G_{FBAR}$ can be calculated by gain-transfer (gain-comparison) method which can be expressed as, $G_{FBAR} = G_R + \log_{10}(P_{FBAR}/P_R)$, where $G_R$ is the gain of the reference horn antenna, and $P_{FBAR}$ and $P_R$ are the radiation power of FBAR and reference horn antenna[46]. Given $\log_{10}(P_{FBAR}/P_R) = S_{21,FBAR} - S_{21,R}$, at the resonance frequency $f_{r,FBAR}$, we obtain $G_{FBAR} = -18$ dBi. The ME FBAR antenna is highly anisotropic due to the strong magnetic film shape anisotropy with a high sensitivity for in-plane magnetic fields, and due to the in-plane uniaxial anisotropy with high sensitivity along the magnetic hard axis of the circular resonating magnetic disk. Directivity $D$ of the ME FBAR antenna can therefore be calculated by integrating the magnetic power density as $D = \frac{\int_0^{2\pi}\int_0^\pi\int_0^\infty \rho \sin\theta\sin\phi\partial\theta d\phi d\rho}{\int P\rho d\rho} = 6$ dB, where $P(\rho, \phi, \theta)$ is the magnetic power density in spherical coordinates. Then the ME FBAR antenna efficiency can be calculated as $\xi_{rad} = G_{FBAR}/D = 0.403\%$ with a high gain of $G_{FBAR} = -18$ dBi at the resonance frequency $f_{r,FBAR}$, or $\xi_{rad,corrected} = 0.448\%$ with reflection corrected. The FBAR ME antenna also has a fractional bandwidth $FBW_{FBAR} = \frac{\Delta f}{f_0} = \frac{BW}{f_0} = 0.158\%$ with the measured 3 dB bandwidth $\Delta f = 4$ MHz. The minimum Q-factor of a small antenna is given by $Q = \frac{1}{(k_0 a)^3} + \frac{1}{k_0 a} = 41037$ as dictated by the Chu limit[44], where $k_0 = \frac{2\pi}{\lambda_0}$ is the wave number in free space and $a$ is the smallest imaginary sphere of radius enclosed the entire antenna structure. The maximum fractional bandwidth of this antenna of the ME antenna allowed by Chu's limit is therefore $FBW_{Chu} \approx \frac{VSWR-1}{\xi_{rad,corrected}Q\sqrt{VSWR}} = 0.628\%$, which is still larger than the measured $FBW_{FBAR} = \frac{\Delta f}{f_0} = \frac{BW}{f_0} = 0.158\%$. Therefore, the Chu–Harrington limit has not been surpassed by using the magnetoelectrically coupled FBAR structure. We also estimated the radiation power of the FBAR antenna by using a simple magnetic dipole model for a conceptual understanding. The magnetic dipole moment ($m_0$) can be expressed as $m_0 = M_s \pi r^2 T$, where $M_s$ is the saturation magnetization, and $r$ and $T$ are the radius and thickness of the magnetic disk. Assuming that a typical input power of -20 dBm (or 0.01 mW) is needed to completely switch all magnetic dipole moment for radiation, we obtain a radiation power ($P_d$) to be $2.8 \times 10^{-8}$ W (or 0.28% efficiency), as $P_d = \frac{\mu_0\omega^4 m_0^2}{12\pi c^3}$, where $c$ is the speed of light in vacuum. This estimation indicates that our experimental results are of the correct order-of-magnitude.

**Data availability**. The data that support the findings of this study are available from the corresponding author upon request.

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

## Acknowledgements

We acknowledge J. Hu, M. Liu and Z. Zhou for discussions. T.N. acknowledges L.C. Sun for assistance in graphic design. This work was supported by DARPA through award D15PC00009, the W.M. Keck Foundation, the NSF TANMS ERC Award 1160504, and in part by the AFRL through contract FA8650-14-C-5706. Microfabrication was performed in the George J. Kostas Nanoscale Technology and Manufacturing Research Center.

## Author contributions

T.N. and H.L. initiated the original idea and led all the device modeling, design and measurements with the supervision of N.X.S.; Z.Q., Y.H., Y.G, H.C. fabricated the devices with the supervision of M.R. G.Y., S.W., M.E.M., B.M.H., Z.H., J.G.J. and G.J.B. assisted the simulations. Z.W. and A.M. annealed the samples. A.B. assisted the SEM measurement. N.S., M. L., X.W., R.G., B.C. and J.Z. helped with the test setup. T.N. and H.L. analyzed the data and prepared the manuscript. All authors discussed the results.

## Additional information

**Competing interests:** N.S. and Northeastern University (NU) have research-related financial interests in Winchester Technologies, LLC. The remaining authors declare no competing financial interests.

