## [Peer Review File · Nature Communications]

Reviewers' comments:

Reviewer #1 (Remarks to the Author):

I have reviewed the manuscript focused on acoustically activated antennas. The manuscript contains important information for the community but it focuses on two (in my opinion) somewhat unrelated devices which represents a potential problem. The more important test data on the (FBAR) related device in the paper receives less focus in the manuscript as compared to the (NPR) to magnetic field sensing device. This includes modeling and testing data which is predominantly focused on the NPR device. As such, it is this reviewers opinion that more information is required for this to be published in Nature, at least for the FBAR portion. My suggestion would be to divide it into two separate manuscripts, note both look very impressive. I provide my comments below that require attention in addition to this general comment provided in this opening paragraph. These comments are divided into comments on the manuscript followed by comments on the supplementary section.

1. In the second paragraph of the second page (intro) authors state that ME effect has only been demonstrated at kHz frequencies. This is NOT correct, there are reports available on several experimental ME structures using acoustic waves in the GHz frequency range (SAW based devices) as well as theoretical models that have modeled acoustically activated antennas. These papers must be (briefly) discussed and referenced. Note that the present manuscript is a significant step beyond what is available in the literature though.

2. The discussion of the NPR device suggests this device represents Electro-Magnetic wave coupling but all the comparisons with analysis appear to be focused on models with magnetostatic approximations, i.e. E and H decoupled (see supplementary section 4 & 7). Therefore, this appears to be a magneto static device rather than an antenna that is coupled with EM radiation. Here the important distinction is coupling between electric and magnetic fields in free space. Authors need to discuss this because all of the test/modeling of this device assume magnetostatics while if this was an EM wave device one would have to model the complicated near field interactions.

3. NPR discussion section needs to show which direction the bias field is applied on the device.

4. NPR, Please better define "in-plane" contour mode for NPR. This could be just a simple schematic in one of the figures. Also the inset of contour plots needs to identify which displacement component is being shown and also a coordinate system to orient the reader. e.g. which is the bias magnetic field direction and what values are being presented.

4a. Did authors conduct tests at large applied magnetic fields, i.e. Figure 2. If so did the test data resemble the non-magnetic sample, i.e. it should because sample is saturated and magnetically locked, This information would be useful in the manuscript but NOT required

5. Summary for NPR work. This represents well presented and well documented test data. My only concern on this work is that the 60MHz magnetic oscillations could be energizing the electrodes on the sample rather than oscillating the spin structure in the FeGaB. Assuming this were correct, the reason that the FeGaB sample would be producing larger voltage out as compared to the dummy sample is the presence of the ferromagnetic material FeGaB modifying the local magnetic energy component, i.e. presence of ferromagnetic material amplifies field. This is easy to eliminate if one time gates the receiving signal, i.e. eliminates the EM component from charging the electrodes or coupling with the cables because the mechanical response should trail this response in the time domain. Note this could be one reason why you are still recording a response for the non-magnetic samples. However, authors have provided sufficient information on NPR work that this represents an excellent contribution.

6. The testing on the FBAR sample truly represents an acoustically actuated antenna. However, insufficient details are provided and what is more concerning is that more time/effort is given for the NPR than the more interesting FBAR work. Thus this reviewer feels more information is needed on these FBAR tests. This includes but is not limited to location of the Horn antenna during test. Some additional information that is required is the stress distribution in the FeGaB at the resonance

frequency. If the authors have this they can predict the magnitude of the magnetization change produce in the FeGaB sample. However, this could present a problem, that is for compressive loading I would expect magnetization reorientation to be absent while for tensile loading it would try to reorient but since the sample is symmetric it would be a coin flip in terms of magnetization reorientation. The acoustic community has recognized that a 45 degree orientation is needed to get good coupling between the stress and the biased magnetization (i.e. bias field). However, modeling results would help clarify this comment. Also, if authors calculate the change in magnitude for the FeGaB Magnetization for the applied stress they can do a simple magnetic dipole calculation to see if the far-field value measured is representative of what they have and NOT representative of coupling to ancillary fields generated during test, i.e. ancillary fields could be coming from electronics/wires. If the value the authors are measuring is higher than this presents a problem for the proposed FBAR EM radiation mechanism. Authors would also benefit the community if they created a "dummy" sample as they did for the NPR sample to demonstrate that radiation is due to magnetic reorientation. However, this later can be problematic because you can still have mechanical resonance that produces EM emissions from the cables/electrodes. This reviewer believes that more work needs to focus on the FBAR which seems relatively incomplete compared to the data presented on NPR samples. Finally, authors did not state what the FMR resonance values are for the FBAR, are they at the same frequency as mechanical resonance?

Comments on Supplementary document.

In general insufficient information is provided in this section for someone to repeat the work presented. Below I provide some comments on the missing information but this should not be considered all inconclusive

1. section 1. What thickness film tested, what is the composition, What is the FMR value, i.e. what frequency is resonance? Was film deposited with bias magnetic field for these tests
2. Section 4, Authors need to state what system of equations they are solving NOT just state Multifephsyc V5.1. Note this looks like authors are solving a magnetostatics problem, i.e. NOT coupled with electric field which does NOT represent an antenna but more of a magnetic sensor This is important because for future studies Comsol may change their modules or even go bankrupt which future researchers would NOT be able to know what and how something was modeled. Also authors need to point out this models is NOT for a radiating antenna. Specifically they are not solving dynamic Maxwell equations. Authors also need to comment on the influence that NPR device has on the local magnetic field, i.e. it will enhance the field. I believe the later comment is not as critical because the authors have shown sufficient work that it is convincing that the sample is responding to the near field magnetic field generated for the NPR structure.
3. Section 5. Did the authors time-gate the EM signal to eliminate the EM reception from interdigitated electrodes. That is the EM wave first impinges on the electrodes while the mechanical response is delayed due to the slower moving mechanical structure. Looking at this in the time domain one should be able to separate out these two contributions and then perform the fourier transform.
4. Section 7. See comments related to section 4, i.e more details on the modeling approach and equations used. I am also happy that authors provided information that they are using electrostatics in this system of equations. Authors need to provide $f(H)$ function used for the ferromagnetic phase. Authors should spend more time describing the magnetics approach employed rather than spend much of this section describing the linear piezoelectric and mechanics used. I see that at the end of this section they state they use linear magnetostriction but this should be placed with the discussion of the nonlinear magnetization response.

Reviewer #2 (Remarks to the Author):

This is interesting paper that deals with important issue. It could be published after the authors

remove minor deficiencies that are listed below. These are:

1) Page 4 line 92

Include the reference to the mode of vibrations

2) Page 5 Lines 126 -129

This sentence is confusing and it should be rewritten. The measured ME voltage amplitude of 196 μmicroV is shown in Figure 1f while the sentence might suggest that it is shown in Figure 1g.

3) Lines 129 – 131

This sentence should also be rewritten because it is confusing, especially the ending.

4) Lines 327

I suspected that the authors missed the word "simulation" at the end of the line.

5) Line 329

I would add the word "presented" after word "is"

6) Line 343

Please check formula for directivity

Response to Reviewers:

We thank the reviewers for the comments and interest in our manuscript, which help us improve the manuscript. We have carefully considered these comments and modified the manuscript accordingly. The reviewers' comments are reproduced below, and our responses and specific actions are described in the blue italicized text. We also include a revised manuscript (including the Supplementary Information) documents that contain the editing markups so that our changes are easily identified.

Here we firstly summarize the major changes in the manuscript:

1. We provide more experimental and simulation information on the FBAR antenna. We have designed, fabricated, and performed the electromechanical and antenna tests on a non-magnetic reference or control FBAR device. A good electromechanical resonance (S_{22}) can be observed, while EM wave radiation/receiving (S_{21} and S_{12}) cannot be detected within our experimental limit as shown in the figure below. This draws a sharp contrast to the magnetolectric FBAR antenna, and demonstrates that the magnetolectric effect dominates in the magnetolectric antenna measurement. The detailed information can be found in the revised Figure 3 and a new corresponding paragraph in the manuscript.

2. We simulated the electromechanical behavior of FBAR device. We show the simulated return loss curve (S_{22}) and the out-of-plane displacement of the circular disk at its resonance frequency below. The detailed information can be found in the revised Figure 3 and the Supplementary section 7.

3. We provide additional information on demonstrating the multi-band, single-chip ME resonators and their potential of realizing reconfigurable antenna arrays. Below shows the resonance frequency and quality factor of various devices fabricated on one single wafer. By simulation and device geometry design, we can achieve a very wide frequency band continuously from 60 MHz to 2.5 GHz on one chip. A bank of multi-frequency MEMS resonators can be

connected to a CMOS oscillator circuit for the realization of reconfigurable ME antenna array. The detailed information can be found in the Supplementary section 6.

4. We reorganized the introduction part to include the discussion of a simulation work and some magnetic surface acoustic wave device (and the corresponding references) as suggested by reviewer 1. We added schematic in Figure 2 for better presenting the bias magnetic field direction with respect to the device direction, as also suggested by reviewer 1. We fixed typos and misleading sentences as pointed out by reviewer 2.

Reviewer #1 Comments and Author Response:

I have reviewed the manuscript focused on acoustically activated antennas. The manuscript contains important information for the community but it focuses on two (in my opinion) somewhat unrelated devices which represents a potential problem. The more important test data on the (FBAR) related device in the paper receives less focus in the manuscript as compared to the (NPR) to magnetic field sensing device. This includes modeling and testing data which is predominantly focused on the NPR device. As such, it is this reviewer’s opinion that more information is required for this to be published in Nature, at least for the FBAR portion. My suggestion would be to divide it into two separate manuscripts, note both look very impressive. I provide my comments below that require attention in addition to this general comment provided in this opening paragraph. These comments are divided into comments on the manuscript followed by comments on the supplementary section.

We appreciate the reviewer’s interest and agree that more experimental and theoretical results are needed in the future for acoustically actuated antenna based on FBAR device. We suggest to keep both NPR and FBAR devices in this manuscript due to the following reasons. (1) The demonstrated NPR and FBAR devices are highly related to each other as both devices operate through acoustically actuated magnetoelastic coupling at their electromechanical resonance. (2) We suggest that we keep both devices in the same manuscript to show to readers that the acoustically actuated magnetoelastic antennas can work across a very wide frequency band. And (3) both the NPR and FBAR devices have the same film stacks and were made through the same batch of wafer during the fabrication process. Also due to the wide operation frequency

range of the NPR and FBAR devices, certain test method can only be performed within a certain frequency range; while the FBAR device is more focused on the antenna test.

1. In the second paragraph of the second page (intro) authors state that ME effect has only been demonstrated at kHz frequencies. This is NOT correct, there are reports available on several experimental ME structures using acoustic waves in the GHz frequency range (SAW based devices) as well as theoretical models that have modeled acoustically activated antennas. These papers must be (briefly) discussed and referenced. Note that the present manuscript is a significant step beyond what is available in the literature though.

Thanks for the comment. We have re-organized the introduction part and added a brief discussion on the surface acoustic wave magnetoelectric devices and a paper on the theory work on the related acoustic actuated antenna.

2. The discussion of the NPR device suggests this device represents Electro-Magnetic wave coupling but all the comparisons with analysis appear to be focused on models with magnetostatic approximations, i.e. E and H decoupled (see supplementary section 4 & 7). Therefore, this appears to be a magneto static device rather than an antenna that is coupled with EM radiation. Here the important distinction is coupling between electric and magnetic fields in free space. Authors need to discuss this because all of the test/modeling of this device assume magnetostatics while if this was an EM wave device one would have to model the complicated near field interactions.

This is a good point, which we very much wish to do. It is very challenging to do 3-D model with complete dynamic Maxwell equations. Scientists at UCLA have published 1-D simple ME antenna structure, which is now added into the reference. EMW coupling in 3D model simulation has not been demonstrated as it is very challenging, which is also beyond the scope for this paper but will be one of our focus in the future. However, in the demonstration of NPR, we excited the device by RF coil instead of the antenna. It is suitable that we only apply AC magnetic field and use magnetostatic approximation to simulate the induced voltage in COMSOL. NPR is more a demonstration of strong ME Coupling and self-biased operation. In the COMSOL simulation of FBAR device, we only demonstrated the resonance mode and displacement instead of the magnetization performance, which the Piezoelectric model is reliable and widely used in COMSOL.

3. NPR discussion section needs to show which direction the bias field is applied on the device.

Thanks for the comment. We have pointed out the direction of the bias magnetic field in the second paragraph on page 7 in our original manuscript. We have also added a small schematic as an inset in Fig. 2b to highlight this bias magnetic field direction, and a coordinate system as the reviewer suggested.

4. NPR, Please better define "in-plane" contour mode for NPR. This could be just a simple schematic in one of the figures. Also the inset of contour plots needs to identify which

displacement component is being shown and also a coordinate system to orient the reader. e.g. which is the bias magnetic field direction and what values are being presented.

Thanks for the comment. The coordinate system and a description of the contour mode plots (Fig. 1d and g insets) have been added as suggested.

4a. Did authors conduct tests at large applied magnetic fields, i.e. Figure 2. If so did the test data resemble the non-magnetic sample, i.e. it should because sample is saturated and magnetically locked, This information would be useful in the manuscript but NOT required

This is another very good point. We agree that the ME voltage tests at high field could be another evidence to support our key findings. We did do ME coupling coefficient measurements at different bias magnetic fields of the NPR device. From this chart, we suspect that a very large bias field will be needed for completely magnetically saturate the NPR antenna with a small magnetic island which are typically much magnetically harder from our test results and from other published data, while our home-made electromagnet on the probe station cannot provide such a large magnetic field to fully saturate the ME antennas.

5. Summary for NPR work. This represents well-presented and well documented test data. My only concern on this work is that the 60MHz magnetic oscillations could be energizing the electrodes on the sample rather than oscillating the spin structure in the FeGaB. Assuming this were correct, the reason that the FeGaB sample would be producing larger voltage out as compared to the dummy sample is the presence of the ferromagnetic material FeGaB modifying the local magnetic energy component, i.e. presence of ferromagnetic material amplifies field. This is easy to eliminate if one time gates the receiving signal, i.e. eliminates the EM component from charging the electrodes or coupling with the cables because the mechanical response should trail this response in the time domain. Note this could be one reason why you are still recording a response for the non-magnetic samples. However, authors have provided sufficient information on NPR work that this represents an excellent contribution.

We agree that the inductive coupling between the EM wave and the electrodes or even the cables is inevitable in such a measurement as we already highlighted in the manuscript. There are two possible coupling mechanisms, (1) ME coupling through the heterostructure, and (2) magnetic film amplifying the ground loop inductive coupling which is minimal as the magnetic film has a near unity relative permeability along its out of plane direction. We believe that the ME coupling effect is much effective than the inductive coupling in the device based on ME heterostructure. Comparing the ME response of the magnetic device (Fig. 1g) with that of the non-magnetic control device (Fig. 1h), not only the amplitude of the ME voltage peak but also the line shape of the spectrum is different. In particular, the magnetic sample shows a symmetric line shape which is consistent with other induced ME voltage spectrum reported at low-frequency (several kHz). While the non-magnetic sample exhibits an antisymmetric line shape, which is very similar to the admittance amplitude curve of that device (Fig. 2e). This observation can be attributed to the inductive coupling effect. Assuming the existence ferromagnetic material could somehow amplify the induced voltage. However, this cannot explain the different line shape we observed for magnetic and non-magnetic devices. And if the inductive coupling dominates, one would see an

antisymmetric spectrum but with larger amplitude for the magnetic device. Thus, from the symmetric behavior of the magnetic device, we concluded that the ME coupling effect dominates. We also agree that a measurement in time domain could be interesting and useful to decouple the ME and inductive coupling effects. But at this stage, we believe a full measurement and discussion on time domain is beyond the scope of the present work.

6. The testing on the FBAR sample truly represents an acoustically actuated antenna. However, insufficient details are provided and what is more concerning is that more time/effort is given for the NPR than the more interesting FBAR work. Thus this reviewer feels more information is needed on these FBAR tests. This includes but is not limited to location of the Horn antenna during test. Some additional information that is required is the stress distribution in the FeGaB at the resonance frequency. If the authors have this they can predict the magnitude of the magnetization change produce in the FeGaB sample. However, this could present a problem that is for compressive loading I would expect magnetization reorientation to be absent while for tensile loading it would try to reorient but since the sample is symmetric it would be a coin flip in terms of magnetization reorientation. The acoustic community has recognized that a 45 degree orientation is needed to get good coupling between the stress and the biased magnetization (i.e. bias field). However, modeling results would help clarify this comment. Also, if authors calculate the change in magnitude for the FeGaB Magnetization for the applied stress they can do a simple magnetic dipole calculation to see if the far-field value measured is representative of what they have and NOT representative of coupling to ancillary fields generated during test, i.e. ancillary fields could be coming from electronics/wires. If the value the authors are measuring is higher than this presents a problem for the proposed FBAR EM radiation mechanism. Authors would also benefit the community if they created a "dummy" sample as they did for the NPR sample to demonstrate that radiation is due to magnetic reorientation. However, this later can be problematic because you can still have mechanical resonance that produces EM emissions from the cables/electrodes. This reviewer believes that more work needs to focus on the FBAR which seems relatively incomplete compared to the data presented on NPR samples. Finally, authors did not state what the FMR resonance values are for the FBAR, are they at the same frequency as mechanical resonance?

This is a good point, and we thank the reviewer's comments. We did provide information regarding different location and rotation of horn antenna in the manuscript (Figure 4). We understand that most of the reviewer's focus is on the modeling. But at this stage, dynamic Maxwell equations or time domain modelling for magnetization reorientation in 3D magnetolectric antenna has not been available, and is beyond the scope of the present work. In response to reviewer's comment, we have designed, fabricated and tested a non-magnetic reference of control antenna for comparison with ME antenna showing that the ME coupling effect dominates the antenna performance comparing with other factors that the reviewer suspected such as the coupling with the ground loop or cables during the test. We believe this is related to the uniaxial magnetic anisotropy. (1) The Young's modulus is highly anisotropic in the plane of the FBAR device, with a weak and field dependent Young's modulus along the magnetic

hard axis direction (i.e. ΔE effect of the FeGaB is as large as 160 GPa along the magnetic hard axis only from our most recent data, but we cannot implement such an anisotropic Young's modulus in our simulation tool), and (2) it is hard to apply a bias field to saturate the ME antenna as the magnetic disk is small and a very large saturation field is needed, as discussed above. The ferromagnetic resonance (FMR) is well defined with a FMR frequency of about 1.6GHz for thin full film; however, the FMR for magnetic small island may occur at much higher frequencies than 1.6 GHz, and not as well defined as full films. We understand that the performance will be maximum while the FMR matches the mechanical resonance according to the scientists in UCLA. However, it is very hard to measure and define the FMR for a small island of FeGaB on a NEMS device, which will be much higher comparing with a full film from our experience.

Comments on Supplementary document.

In general insufficient information is provided in this section for someone to repeat the work presented. Below I provide some comments on the missing information but this should not be considered all inconclusive

1. section 1. What thickness film tested, what is the composition, What is the FMR value, i.e. what frequency is resonance? Was film deposited with bias magnetic field for these tests

The information has been added to the SI document as suggested.

2. Section 4, Authors need to state what system of equations they are solving NOT just state Multiphysics V5.1. Note this looks like authors are solving a magnetostatics problem, i.e. NOT coupled with electric field which does NOT represent an antenna but more of a magnetic sensor This is important because for future studies Comsol may change their modules or even go bankrupt which future researchers would NOT be able to know what and how something was modeled. Also authors need to point out this models is NOT for a radiating antenna. Specifically they are not solving dynamic Maxwell equations. Authors also need to comment on the influence that NPR device has on the local magnetic field, i.e. it will enhance the field. I believe the later comment is not as critical because the authors have shown sufficient work that it is convincing that the sample is responding to the near field magnetic field generated for the NPR structure.

As we mentioned before, it is very challenging to do 3-D modeling with complete dynamic Maxwell equations. EMW coupling in 3D model simulation has not been demonstrated as it is very challenging, which is also beyond the scope for this paper but will be one of our focus in the future. For NPR simulation, we apply AC magnetic field and use magnetostatic approximation to simulate the induced voltage in COMSOL which is exactly what we did for the measurement. We did indicate that we applied existing modules in COMSOL which are magnetic fields, solid mechanics, and electrostatics modules. The equations are provided in the Supplementary Information. NPR is more like a demonstration for capability of ME Coupling and self-biasing. In FBAR simulation section, we only demonstrated the resonance mode and displacement

instead of the magnetization performance which the Piezoelectric model is reliable and widely used in COMSOL. We didn't discuss or claim any radiating capability of our FBAR modeling, and we have updated the manuscript and pointed it out.

3. Section 5. Did the authors time-gate the EM signal to eliminate the EM reception from interdigitated electrodes. That is the EM wave first impinges on the electrodes while the mechanical response is delayed due to the slower moving mechanical structure. Looking at this in the time domain one should be able to separate out these two contributions and then perform the Fourier transform.

As we respond in this reviewer's question No.5 above, we believe that time domain measurement is beyond the scope of the present work. We appreciate that the reviewer raised up a very interesting point which requires further work to address and is indeed being pursued.

4. Section 7. See comments related to section 4, i.e. more details on the modeling approach and equations used. I am also happy that authors provided information that they are using electrostatics in this system of equations. Authors need to provide $f(H)$ function used for the ferromagnetic phase. Authors should spend more time describing the magnetic approach employed rather than spend much of this section describing the linear piezoelectric and mechanics used. I see that at the end of this section they state they use linear magnetostriction but this should be placed with the discussion of the nonlinear magnetization response.

Reviewer #2 Comments and Author Response:

This is an interesting paper that deals with an important issue. It could be published after the authors remove minor deficiencies that are listed below. These are:

1) Page 4 line 92

Include the reference to the mode of vibrations

2) Page 5 Lines 126 -129

This sentence is confusing and it should be rewritten. The measured ME voltage amplitude of 196 μV is shown in Figure 1f while the sentence might suggest that it is shown in Figure 1g.

3) Lines 129 – 131

This sentence should also be rewritten because it is confusing, especially the ending.

4) Lines 327

I suspected that the authors missed the word "simulation" at the end of the line.

5) Line 329

I would add the word "presented" after word "is"

6) Line 343

We appreciate the reviewer's comments. We have fixed all typos and misleading sentences where indicated.

Reviewer #3 Comments and Author Response:

1. [REVIEWER COMMENTS REDACTED]

We appreciate the reviewer's comments. We believe our demonstrated ME antennas (both NPR and FBAR) are significantly and fundamentally different than the conventional RF magnetic field actuator. RF magnetic field actuators often refer to the devices that can be actuated by the bending moment from the alternating current induced by a RF magnetic field. The electrical readout of this kind of device relies on additional sensing coils. Long story short, conventional actuators have nothing to do with the magnetoelectric coupling between the piezoelectric and ferromagnetic phases in one resonator. In this work, we show strong evidences that the ME coupling effect dominates in the ME antennas and produces much higher efficiency than inductive coupling effect. In particular, for the ME FBAR device, the antenna measurements were performed with a horn antenna. Clear resonance peaks can be obtained for both antenna receiving and radiation. We also provided the radiation patterns by rotating the axis of the horn antenna in 3 different orthogonal directions. In the revised manuscript, we directly show that the antenna radiation and receiving efficiency are much lower for the non-magnetic reference FBAR device, suggesting that the magnetoelectric coupling dominates in the ME FBAR antenna. This is not related to any study of RF magnetic field actuator or electromagnetic actuator.

2. [REVIEWER COMMENTS REDACTED]

We respectively disagree with the reviewer's comment. Magnetoelectric antennas have been the research focus of our current NSF Engineering Research Center TANMS (Translational Applications of Nanoscale Multiferroic Systems), and the focus of multiple current DARPA and NSF projects which we are working on. Compared to same-size conventional antenna designs, these ME antennas show 20~50dB gain enhancement. This manuscript represented the best results from these efforts on ME antennas, which have attracted more and more interests from many major DoD contractors. These are clear signs that magnetoelectric (ME) antennas have their unique advantages that have a strong potential for practical applications with competitive specifications.

3. [REVIEWER COMMENTS REDACTED]

We have provided such a comparison in the original version of the manuscript. The detailed information can be found in the Supplementary Section 8. In a small loop antenna with the same size as the ground loop of the ME FBAR and with good impedance matching, the antenna gain is 50 dB lower than that of the FBAR ME antenna. In the revised manuscript, we also added the antenna test results of a non-magnetic reference FBAR antenna. There is a sharp difference

between the ME antenna and the reference antenna, where no evident S_{21} and S_{12} resonance peak can be observed in the horn antenna measurements for the reference antenna, except a very weak peak at 2.50 GHz with a peak amplitude merely above the noise level. This suggests that a similar non-magnetic control device without the magnetolectric coupling would have very poor antenna properties.

4. [REVIEWER COMMENTS REDACTED]

One purpose of this paper is to demonstrate the concept of ME antennas through ME coupling effect, not for a specific application or frequency need. These ME antennas can be applied to devices with different modes of vibration in a very large frequency range. As shown above in the beginning of our response to reviewers, we have designed, fabricated and tested many different magnetolectric antennas. By simulation and device geometry design, we can achieve a very wide frequency band continuously from 60 MHz to 2.5 GHz on one chip. A bank of multi-frequency MEMS resonators can be connected to a CMOS oscillator circuit for the realization of reconfigurable ME antenna array. The detailed information can be found in the Supplementary section 6. The fundamental frequency of NPR device, which operates in the contour mode, can be set by the lateral dimension of the structure. Such dimensions can be defined directly at the CAD layout level. Potentially, antennas based on NPR in one wafer could cover the frequencies from tens of MHz to 1 GHz. Then, the demonstrated FBAR device uses this concept and further expand the resonance frequency to several GHz by a device geometric design with different modes of vibration. In the revised manuscript, we showed the resonance frequency and quality factor of various devices fabricated on one single wafer. By simulation and device geometry design, we can achieve a very wide frequency band continuously from 60 MHz to 2.5 GHz on one chip.

Reviewers' comments:

Reviewer #1 (Remarks to the Author):

I would like to have seen the authors spend more time on validating the FBAR antenna approach in this manuscript. Specifically conducting a basic piezomagnetic model to predict what the magnetizations changes are in the FBAR devices could be accomplished. Specifically they have the stresses in the FBAR, thus they should be able to use a 1D constitutive relation to predict what M changes are occurring from these stresses. Having this as a baseline, authors could then use a magnetic dipole model to predict the radiation at frequency of a magnetic dipole and see if the measured response is comparable to that of the predicted magnetic dipole model.

Reviewer #2 (Remarks to the Author):

Reviewer #3 (Remarks to the Author):

Many thanks for the reply to my comments on the possibility of using the proposed system as antennas. The antenna by virtue of its definition is a transducer designed to transmit or receive electromagnetic waves. When it has below -18dBi path gain as indicated in the paper for the proposed ME antenna, engineers will find it difficult to use as an efficient EM wave radiation device. As the authors indicated, there exists fundamental limits for electrically small antennas. However, one must bear it in the mind that the Chu-Harrington limit was derived from a passive structure. There have been a rich collection of literature demonstrating how such a limit can be overcome by using active inclusions such as non-foster circuits. This is also a topical research in 1950s, which has attracted recently renewed interests in the antenna community. Yes indeed, the antenna size can be further reduced and impedance match but the overall system noise figure will increase, hence will not increase the system bandwidth/capacity i.e. for communication applications.

The proposed antenna, once termed as "Strain powered antennas" by John P. Domann and Greg P. Carman (Citation: Journal of Applied Physics 121, 044905 (2017); doi: 10.1063/1.4975030), it serves the same purpose for antenna size reduction. In the reference "Bulk Acoustic Wave-Mediated Multiferroic Antennas: Architecture and Performance Bound", the authors are very cautious about the claim which the proposed antenna will overcome the Chu's limit. For this paper, the authors may have done an excellent job in developing efficient FBARs. However, its analysis of use in antenna design is incomplete and not rigorous. I have some specific questions regarding the antenna design:

1. what is the mechanism for the antenna size reduction? Is it because of increase in permittivity and permeability by the induce of acoustic waves. If so, can these values be probably measured and used in the simulation of antennas as effective medium parameters? What is the loss associated with the increase in permittivity and permeability? Are these two values identical or close to each other as indicated in magneto-dielectrics?
2. all dimensions of antenna designs shall be provided in details, these must include a detailed drawing of FBAR, metallic electrodes and their interfaces as well as other substrate materials, this will help readers to understand whether or not metallic pads act as dipole antennas?

3. figure 3 presents both return loss and path gain for the proposed ME antenna. It is rather odd to see that despite of good impedance matching for non-magnetic biasing, the path gain has been reduced by almost 20dB, where is the energy lost?

4. it is very well known that accurately measuring electrically small antennas is not trivial, as potentially, current leakages from coaxial cables which connects the DUT and PNA will cause even greater radiation than those from antennas. A rigorous approach is to use fibre-optical system to feed the antenna. The authors are encouraged to analyse measurement errors in the paper.

5. the authors must include "Strain powered antennas" by John P. Domann and Greg P. Carman in the reference.

In summary, I appreciate that the authors have demonstrated a new FBAR design and fabrication. Its use in antennas is not very well presented, in particular, the claim for overcoming the Chu's limit is too harsh. The authors may rewrite some parts of this paper, emphasizing its use as multi-functional RF devices other than electrically small antennas.

Response to Reviewers:

We appreciate the reviewers' feedbacks on our revised manuscript (NCOMMS-16-27516A), which help us further improve the manuscript. We have carefully considered these comments and modified the manuscript accordingly. The reviewers' feedbacks are copied below, and our responses and specific actions are described in the blue italicized text. We also include the revised manuscript and Supplementary Information that contain the editing markups so that our changes are easily identified.

Reviewer #1 Comments and Author Response:

I would like to have seen the authors spend more time on validating the FBAR antenna approach in this manuscript. Specifically conducting a basic piezomagnetic model to predict what the magnetizations changes are in the FBAR devices could be accomplished. Specifically they have the stresses in the FBAR, thus they should be able to use a 1D constitutive relation to predict what M changes are occurring from these stresses. Having this as a baseline, authors could then use a magnetic dipole model to predict the radiation at frequency of a magnetic dipole and see if the measured response is comparable to that of the predicted magnetic dipole model.

We thank the reviewer for the suggestion. At the same time, we iterate that a quantitative analysis of such a complex system as requested by the reviewer is beyond the scope of this manuscript. The 1D multiscale finite-difference time domain (FDTD) theoretical analysis of a magnetostrictive/piezoelectric bulk acoustic wave resonator (BAW) has been published and cited in our manuscript (Yao et al. IEEE Trans. Antennas Propag. 63, 3335–3344 2015). In a similar system, the proposed strain-mediated antenna in Yao's paper shows the capability to radiate watts of power. Experimentally, as responded in our first-round response letter and the revised manuscript, we have presented the non-magnetic reference FBAR device which draws a sharp contrast to the magnetoelectric FBAR antenna. We then validated that the ME coupling effect is dominating in the ME FBAR antenna as the EM wave radiation/receiving efficiency in the reference FBAR device is extremely weak.

By using the simple magnetic dipole model to predict the radiation as the reviewer suggested, we have calculated the radiation power to be $2.8 \times 10^{-8} \text{W}$ assuming that all magnetic dipole moment is switched at the electromechanical resonance. If we assume a typical input power of -20dBm (or 0.01mW) is needed to completely switch all magnetic dipole moment for radiation, we get 2.8×10^{-3} , or 0.28% efficiency, which is close to what we got in the manuscript. However, as we mentioned in our first-round response letter, it is not trivial to estimate the radiation of such a 3D device. And by using the 1D model, one may not be able to capture the real physics of our device which contains many boundary conditions and anisotropic materials parameters. For example, the magnetic layer FeGaB thin film in the ME antenna shows a highly anisotropic Young's modulus with a ΔE of 160 GPa along the in-plane magnetic hard axis direction, which is very hard to incorporate into any existing model. Therefore, we have been focusing on the experimental part in the manuscript.

Reviewer #3 Comments and Author Response:

Many thanks for the reply to my comments on the possibility of using the proposed system as antennas. The antenna by virtue of its definition is a transducer designed to transmit or receive electromagnetic waves. When it has below -18dBi path gain as indicated in the paper for the proposed ME antenna, engineers will find it difficult to use as an efficient EM wave radiation device. As the authors indicated, there exists fundamental limits for electrically small antennas. However, one must bear it in the mind that the Chu-Harrington limit was derived from a passive structure. There have been a rich collection of literature demonstrating how such a limit can be overcome by using active inclusions such as non-foster circuits. This is also a topical research in 1950s, which has attracted recently renewed interests in the antenna community. Yes indeed, the antenna size can be further reduced and impedance match but the overall system noise figure will increase, hence will not increase the system bandwidth/capacity i.e. for communication applications. The proposed antenna, once termed as "Strain powered antennas" by John P. Domann and Greg P. Carman (Citation: Journal of Applied Physics 121, 044905 (2017); doi: 10.1063/1.4975030), it serves the same purpose for antenna size reduction. In the reference "Bulk Acoustic Wave-Mediated Multiferroic Antennas: Architecture and Performance Bound", the authors are very cautious about the claim which the proposed antenna will overcome the Chu's limit. For this paper, the authors may have done an excellent job in developing efficient FBARs. However, its analysis of use in antenna design is incomplete and not rigorous. I have some specific questions regarding the antenna design:

We appreciate the reviewer's comments. We would like to clarify that our ME antenna is a passive device, not active device as the reviewer interpreted. There is no connection to an external power source, either DC and AC voltage or magnetic field during the measurement. Our ME antenna shows a higher gain and better impedance matching than an electrically small antenna with the same size (small loop antenna in the manuscript). Therefore, we believe the Chu-Harrington limit is appropriate to be discussed in the manuscript and the limit has not been surpassed using magnetoelectrically coupled FBAR structure. The detail of the calculation can be found in Method section of the manuscript.

1. what is the mechanism for the antenna size reduction? Is it because of increase in permittivity and permeability by the induce of acoustic waves. If so, can these values be probably measured and used in the simulation of antennas as effective medium parameters? What is the loss associated with the increase in permittivity and permeability? Are these two values identical or close to each other as indicated in magneto-dielectrics?

As discussed in our manuscript, and also proposed by another theoretical paper (Z. Yao et al. IEEE Trans. Antennas Propag. 63, 3335–3344 2015) in our reference, the mechanism for the ME antenna operation and miniaturization is due to the magnetoelectric effect at the acoustic resonance frequency or electromechanical resonance frequency. Since the acoustic wavelength is much less than that of the electromagnetic wave resonance, these ME antennas are much smaller than state of the art compact antennas.

Size miniaturization of ME FBAR antennas is not due to the high permeability or high permittivity, different from conventional magnetodielectric antenna approaches (R. Petrov et al. Electron. Lett. 44 8 506 2008, G. Yang et al. IEEE Trans. Mag. 44.11 3091-3094 2008). The loss mechanism of ME antennas is quite different from conventional antennas as the mechanical resistance is dominating the loss of ME antennas. And the mechanical resistance is not directly related to the loss tangent of the piezomagnetic or piezoelectric phases of the ME antennas.

2. all dimensions of antenna designs shall be provided in detail, these must include a detailed drawing of FBAR, metallic electrodes and their interfaces as well as other substrate materials, this will help readers to understand whether or not metallic pads act as dipole antennas?

We thank the reviewer's suggestion. The dimensions and 3D structure of FBAR antenna are provided in the following figure. We have also added this figure to the Supplementary Information.

3. figure 3 presents both return loss and path gain for the proposed ME antenna. It is rather odd to see that despite of good impedance matching for non-magnetic biasing, the path gain has been reduced by almost 20dB, where is the energy lost?

As described in (2) above, the loss mechanism of ME antennas is dominated by the mechanical resistance $R_{mechanical}$ related to the different mechanical damping mechanisms of the magnetic and piezoelectric phases, which is much larger than the radiation resistance $R_{radiation}$. The impedance matching is therefore dominated by $R_{mechanical}$, not $R_{radiation}$. Therefore, impedance matching is no longer directly related to the radiation efficiency of ME antennas.

4. it is very well known that accurately measuring electrically small antennas is not trivial, as potentially, current leakages from coaxial cables which connects the DUT and PNA will cause even greater radiation than those from antennas. A rigorous approach is to use fiber-optical system to feed the antenna. The authors are encouraged to analyze measurement errors in the paper.

This is a good point, and we agree with the comments above. As we have addressed in our first response letter, the spurious effects, such as the inductive coupling between the device ground loop and cables, have been considered. We tested a non-magnetic reference FBAR device which shows a comparable electro-mechanical property as the ME FBAR device. Any inductive coupling would also be picked up by the non-magnetic reference sample. However, the transmitting/receiving signals for magnetic devices are significantly larger than the non-magnetic ones.

5. the authors must include "Strain powered antennas" by John P. Domann and Greg P. Carman in the reference.

We thank the reviewer's suggestion. We have now included this in the reference.

In summary, I appreciate that the authors have demonstrated a new FBAR design and fabrication. Its use in antennas is not very well presented, in particular, the claim for overcoming the Chu's limit is too harsh. The authors may rewrite some parts of this paper, emphasizing its use as multi-functional RF devices other than electrically small antennas.

Again, we want to point out that our device is purely passive and is within the Chu limit as we have made clear above already. In the original manuscript, we discussed that part in the Method. We have updated the manuscript (the second to last paragraph in the main manuscript) to make it clearer that our ME antennas are passive and are within Chu limit.

From what we have presented in the manuscript, we do believe that this work represents ground-breaking demonstration on a new approach to orders of magnitude reduced antenna size for both receiving and transmitting electromagnetic waves. Therefore, we respectfully disagree with the reviewer's comment, and have to insist that we call these NEMS ME resonators as ME antennas.

REVIEWERS' COMMENTS:

Reviewer #1 (Remarks to the Author):

I see the authors made the calculations of the magnetic dipole but did not include this in the manuscript. I think this needs to be included with a brief discussion. It will help others working in this area to get a sense of how to predict the response even though that this is a simplified approach to a complex problem

Reviewer #3 (Remarks to the Author):

I am happy with the revision made by authors and the paper is now recommended for publication.

Response to Reviewers:

We appreciate the reviewers' feedbacks on our revised manuscript (NCOMMS-16-27516B), which help us further improve the manuscript.

Reviewer #1 Comments and Author Response:

I see the authors made the calculations of the magnetic dipole but did not include this in the manuscript. I think this needs to be included with a brief discussion. It will help others working in this area to get a sense of how to predict the response even though that this is a simplified approach to a complex problem.

We thank the reviewer for the suggestion. We have now included the radiation power estimation based on a simple magnetic dipole model into Method.

Reviewer #3 Comments and Author Response:

I am happy with the revision made by authors and the paper is now recommended for publication.

We appreciate the reviewer's recommendation.